# Changes in the salt content of packaged foods sold in supermarkets between 2015–2020 in the United Kingdom: A repeated cross-sectional study

**Lauren K. Bandy**[1,2]*, **Sven Hollowell**[1], **Susan A. Jebb**[2], **Peter Scarborough**[1]

**1** Nuffield Department of Population Health, University of Oxford, Oxford, United Kingdom, **2** Nuffield Department of Primary Care Health Sciences, University of Oxford, Oxford, United Kingdom

* lauren.bandy@phc.ox.ac.uk

## Abstract

### Background

Excess consumption of salt is linked to an increased risk of hypertension and cardiovascular disease. The United Kingdom has had a comprehensive salt reduction programme since 2003, setting a series of progressively lower, product-specific reformulation targets for the food industry, combined with advice to consumers to reduce salt. The aim of this study was to assess the changes in the sales-weighted mean salt content of grocery foods sold through retail between 2015 and 2020 by category and company.

### Methods and findings

Information for products, including salt content (g/100 g), was collected online from retailer websites for 6 consecutive years (2015 to 2020) and was matched with brand-level retail sales data from Euromonitor for 395 brands. The sales-weighted mean salt content and total volume of salt sold were calculated by category and company. The mean salt content of included foods fell by 0.05 g/100 g, from 1.04 g/100 g in 2015 to 0.90 g/100 g in 2020, equivalent to −4.2% ($p = 0.13$). The categories with the highest salt content in 2020 were savoury snacks (1.6 g/100 g) and cheese (1.6 g/100 g), and the categories that saw the greatest reductions in mean salt content over time were breakfast cereals (−16.0%, $p = 0.65$); processed beans, potatoes, and vegetables (−10.6%, $p = 0.11$); and meat, seafood, and alternatives (−9.2%, $p = 0.56$). The total volume of salt sold fell from 2.41 g per person per day to 2.25 g per person per day, a reduction of 0.16 g or 6.7% ($p = 0.54$). The majority (63%) of this decrease was attributable to changes in mean salt content, with the remaining 37% accounted for by reductions in sales. Across the top 5 companies in each of 9 categories, the volume of salt sold decreased in 26 and increased in 19 cases. This study is limited by its exclusion of foods purchased out of the home, including at restaurants, cafes, and takeaways. It also does not include salt added at the table, or that naturally occurring in foods, meaning the findings underrepresent the population's total salt intake. The assumption was also made that the products matched with the sales data were entirely

**Data Availability Statement:** This study used data from two commercial sources. The sales data was accessed under licence from Euromonitor

International (https://www.euromonitor.com/packaged-food) via the Bodleian Library, University of Oxford, using Euromonitor's database portal Passport GMID. The product information dataset, including nutrition composition data, was purchased for the purpose of the lead author's DPhil research project from Edge by Ascential (https://www.ascentialedge.com/our-solutions). Due to licencing restrictions, the Euromonitor and Edge by Ascential datasets can only be requested under licence for the purpose of verification and replication of study's findings via the research group's Data Access Committee (contact: Trisha Gordon foodDBaccess@ndph.ox.ac.uk). Further use of these datasets must be negotiated with the data owners (Euromonitor contact: Ashton Moses - passport.support@euromonitor.com, Edge by Ascential contact: David Beech - info@ascentialedge.com).

**Funding:** LB was funded by Nuffield Department of Population Health, University of Oxford and the National Institute for Health Research (NIHR) Applied Research Collaboration (ARC) Oxford and Thames Valley. LB, SH, SJ and PS were also funded by the NIHR Oxford Biomedical Research Centre (BRC) Obesity, Diet and Lifestyle Theme. The funders had no role in study design, data collection and analysis, decision to publish, or preparation of the manuscript.

**Competing interests:** The authors have declared that no competing interests exist.

representative of the brand, which may not be the case if products are sold exclusively in convenience stores or markets, which are not included in this database.

## Conclusions

There has been a small decline in the salt content of foods and total volume of salt sold between 2015 and 2020, but observed changes were not statistically significant so could be due to random variations over time. We suggest that mandatory reporting of salt sales by large food companies would increase the transparency of how individual businesses are progressing towards the salt reduction targets.

## Author summary

### Why was this study done?

- Excess salt consumption leads to increased risk of high blood pressure and heart disease. Reducing salt intake across the population is a public health priority.

- Voluntary salt reduction targets for the food industry in the United Kingdom cover a wide range of foods, such as bread, snacks, sauces, and ready meals.

- This study examines how the sales-weighted mean salt content of the 9 grocery food categories that contribute the most to adults' salt intake in the UK has changed between 2015 and 2020, and how it has changed by individual company.

### What did the researchers find?

- Nine grocery food categories that contribute the most to adult salt intake in the UK were included in this study.

- The mean salt content of these foods fell by 5%, from 1.04 g/100 g to 0.99 g/100 g, although the results are not statistically significant.

- The biggest reductions were seen in breakfast cereals (−16%) and processed beans, potatoes, and vegetables (−11%), but there was no change for bread (−2%) and ready meals (+1%). None of these changes were statistically significant.

### What do the findings mean?

- There has been little change in the mean salt content and total volume of salt sold from these foods.

- Overall, progress in salt reduction has stalled. Additional policy measures might be needed to further reduce the salt content of foods, such as mandatory reporting of salt sales by manufacturers, which might improve transparency and simulate further progress.

## Introduction

Excess consumption of salt leads to an increased risk of hypertension [1,2] and is linked to 3 million deaths a year globally, mostly from strokes and cardiovascular disease [3]. There is strong evidence from randomised control trials that shows reducing dietary salt intake leads to reductions in both systolic and diastolic blood pressure [4–6]. Reducing salt intake is a public health priority and has been identified as one of the most cost-effective measures a country can take to improve population health outcomes [7]. Public health surveillance of salt intake, reducing salt content of foods, labelling and marketing standards, and increasing public awareness of the relationships between salt intake and health have all been shown to reduce population salt intake [8]. WHO recommends salt intake does not exceed 5 g/day [9], although more recently, questions have been raised about the negative consequences of low dietary sodium intake on cardiovascular health, particularly for high-risk groups such as those with diabetes [10].

The United Kingdom has one of the best-known salt reduction programmes globally, aiming to achieve a population salt intake of less than 6 g/day. The UK Government first published a comprehensive salt reduction programme in 2003, setting a series of voluntary, progressively lower, product-specific reformulation targets for the food industry to achieve in 4 years, combined with advice to consumers to reduce salt. Subsequent targets were set for 2009, 2011, 2014, and 2017. The most recent salt reduction targets have been set for 2024 and include 84 specific food groups that contribute the most to salt intake, and cover products sold for consumption both in and out of the home [11].

Previous studies have reported some success in reducing the salt content of foods in the UK. A study that analysed the sodium content of 47,000 foods purchased in 18,000 households in the UK between 2006 and 2011 found that the mean content fell by 7% overall, concluding that the voluntary reduction targets had delivered a moderate reduction in the salt content of foods over the time period studied [12]. A cross-sectional study examining the salt content of foods reported in the National Diet and Nutrition Survey (NDNS) found that the sodium density had reduced by 17% in 2016 to 2017 compared to 2008 to 2009 [13]. In 2020, Public Health England (PHE) has published a report on the salt content of foods in relation to the 2017 targets, based on a single year of data. It found that of the foods purchased for at-home consumption, around half of the average salt reduction targets had been met, with 20% of products still exceeding the maximum salt content targets [14]. However, there has been limited reporting of the results by individual companies.

Assessment of salt intake from urinary sodium excretion in adults has shown that estimated mean salt intake reduced by 1.4 g/day between 2003 and 2011, contributing to a reduction of 3.0 mm Hg in systolic blood pressure and a reduction in mortality from stroke and ischaemic heart disease [15]. Interrupted time series models have shown that mean salt intake fell by 0.20 g/day among men and 0.12 g/day among women from 2003 to 2010, but then slowed to 0.11 g/day for men and 0.07 g/day for women between 2011 and 2014 [16]. The most recent data from NDNS' 2018 to 2019 assessment of urinary sodium in adults reported a mean intake of 8.4 g/day [17], with no significant change observed compared to 2014. All this evidence suggests that the population's salt intake has plateaued in more recent years.

Given the patchy progress in reformulation towards the targets across categories reported by PHE and the plateauing of the population's salt intakes, this study set out to assess the changes in the sales-weighted mean salt content of grocery foods in the UK between 2015 and 2020 by category and company. We specifically aimed to assess how industry-led reformulation (salt content) and consumer behaviour (sales) have contributed to estimated changes in salt intake over time.

## Methods

This repeated cross-sectional study was conducted between April 2021 and January 2022 and was not part of any preplanned analyses and has no assisted protocol. The sensitivity analysis was added during the peer review process, and this study is reported as per the Strengthening the Reporting of Observational Studies in Epidemiology (STROBE) guideline (S1 STROBE Checklist).

First, the top categories contributing to adult (19 to 64 years) dietary sodium intake were identified using results of the latest NDNS survey [18], and products in the following 9 categories were included: bread; breakfast cereals; butter and spreads; cheese; meat, seafood, and alternatives; processed beans, potatoes, and vegetables; ready meals, soup, and pizza; sauces and condiments; and savoury snacks. The included subcategories are given in Table 1. Based on the latest NDNS data, 8 of these categories represent an estimated 73% of an adult's dietary sodium intake from processed foods, with no estimate available for ready meals [18].

### Data types and sources

Data on the salt content of foods were acquired from 2 sources, depending on the date. Data for the years 2015 to 2018 were sourced from a commercial third-party supplier, Edge by Ascential (previously known as Brand View). Edge by Ascential collects product information, including nutrient composition data from the websites of 4 leading UK retailers: Asda, Morrisons, Sainsbury's, and Tesco, with 2015 being the earliest data available. All data were collected on the same data (13 December) for each year. For 2018 to 2020, similar data were sourced from foodDB, a research database developed by the University of Oxford [19]. These data were collected in November and December for the same 4 retailers for 2018, 2019, and 2020. For 2018, data from both sources were used in the analysis.

**Table 1. Included sales data categories and estimated contribution to sodium intake according to NDNS [18].**

| Sales data category | Included sales data subcategories | % contribution of category to total sodium intake for adults (19–64) [16] |
|---|---|---|
| Meat, seafood, and alternatives | Chilled and frozen meat substitutes, chilled and frozen processed poultry, chilled and frozen processed red meat, chilled frozen processed seafood, shelf stable meat | 32 |
| Bread | Bread | 15 |
| Processed beans, potatoes, and vegetables | Frozen processed potatoes, shelf stable beans, shelf stable tomatoes, shelf stable vegetables | 9 |
| Sauces and condiments | Barbecue sauces, chili sauces, cooking sauces, dips, ketchup, mayonnaise, mustard, pasta sauces, salad dressings, soy sauces, other sauces | 6 |
| Cheese | Hard cheese, processed cheese, soft cheese | 5 |
| Breakfast cereals | Hot cereals, ready-to-eat cereals | 2 |
| Butter and spreads | Butter and spreads | 2 |
| Savoury snacks | Popcorn, potato crisps, puffed snacks, rice snacks, tortilla chips, vegetable, pulse and bread chips | 2 |
| Ready meals, soup, and pizza | Chilled, shelf stable and frozen ready meals, chilled and frozen pizza, soup | NA |
| TOTAL | | 73 |

The data from each source were combined to form a composition database with the following information: year, product name, brand name, company (manufacturer) name, ingredients, and salt content in g per 100 g. Products that did not have a salt per 100 g value were excluded, as were products that did not contain values for at least 2 other nutrients. No missing data or salt values were imputed. As barcode data were not available, duplicate products found in multiple retailers were removed by filtering on year, category, brand name, product size, kcal per 100 g, and salt per 100 g. An extra analysis was done to test the validation between the 2 sources of composition data (Edge by Ascential and foodDB) for 2018. The sales-weighted mean salt content by category was calculated for each data source, and a Wilcoxon signed-rank test was conducted in R to test for differences between the 2 independent samples, with the unit of analysis being the brand.

There were 374,849 individual products included in the original composition database for all years. A total of 80,960 products were excluded due to missing salt per 100 g values; 205,003 were excluded because they did not belong in one of the included categories (as outlined in Table 1); and 105 were excluded because they were missing values for 3 or more other nutrients. A total of 88,781 products were included in the final composition database for all years.

The existing categories and retailer shelf names in the composition database were used to assign products to the new sales categories (see Table 1) where possible, otherwise, key word searches were conducted on product names to ensure that all products were assigned a sales category. This process was done in Microsoft Excel and R Studio (version 2022.02.3). Canned fish was not included in the Edge by Ascential database and so was excluded. Products with a relatively low salt content, including sweet biscuits, cakes, pastries and plain pasta, rice, and noodles, were excluded. There was no information provided on whether the salt content of dried sauces, gravy, and stock cubes was given per 100 g as sold or as served; therefore, these products were excluded to remove inconsistencies.

## Pairing composition data with sales data

Weighting composition data by sales volume allows the total volume of salt sold to be estimated, which can be used to estimate average salt purchases (as a proxy for consumption) within the population. For this study, brand-level sales data were sourced from Euromonitor International via the Bodleian Library, University of Oxford. Euromonitor is a private market research company that provides retail sales data in value and volume terms and is representative of the whole packaged food market. More details on Euromonitor sales data used in this study are provided elsewhere [20].

Euromonitor only measures sales by brand and so individual product sales data were not available. A brand was defined as a set of products that have the same core name and are manufactured by the same company. For example, the company Kellogg's manufactures multiple brands, including Special K (one brand) and Coco Pops (another brand) breakfast cereals. Within each brand, there may be multiple individual products (e.g., Special K Original and Special K Red Berries). The brand-level sales data were matched with product-level salt content data based on product name, brand name, company name, category, and year. Where brands were matched with more than 1 individual product, mean salt content was calculated. This process is outlined in the flow chart in Fig 1 below. Each year, any new brands that were added to the market place were included, and any brands that were removed were excluded (i.e., this study included all brands, not just those present throughout the whole time series).

Corresponding product information from the composition database could not be found for 58 brands in the sales database, either because these brands were not sold in the 4 retailers included in the database ($n = 18$) or manufactured in the given time period ($n = 40$). These

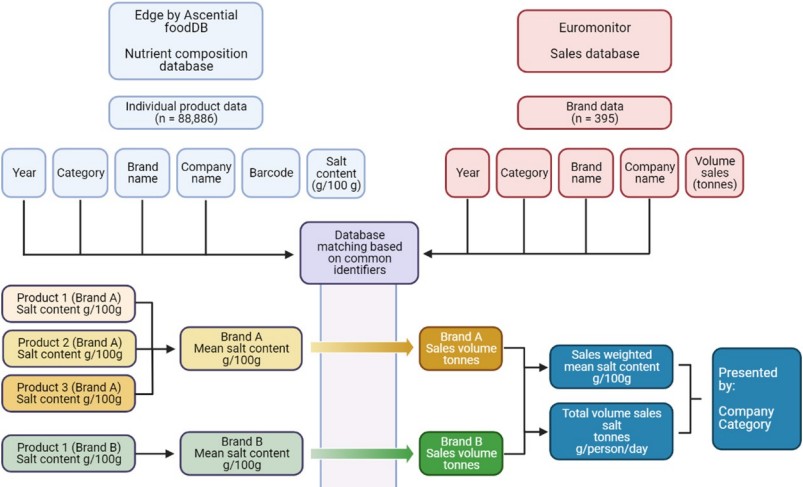

**Fig 1. Data processing and analysis flow chart.**

brands were therefore removed from the sales database (Fig 1). Euromonitor classifies small and local brands under the umbrella of "others," representing 8% of sales overall and ranging from 3% (breakfast cereals) to 24% (savoury snacks) for category-level sales. The remaining products in the composition database that had not been matched to a brand were assigned to "others" within each category, with the mean salt content being weighted by the sales value for "others."

## Data analysis

The unit of analysis was the brand, and the sales-weighted mean salt content (g/100 g) was the primary outcome. Total volume of salt sold was also calculated in tonnes. The category, brand, and company variables in the datasets meant that analyses were stratified by company and category. Change in sales-weighted mean salt content was calculated between 2015 and 2020. In order to put the results in context of dietary recommendations, results for total volume of salt sold are presented in grams per person per day instead of tonnes. These values were calculated by dividing tonnes by population size using annual population estimates from the Office for National Statistics [21] and 365 to give daily rather than annual figures.

The change in total volume of salt sold was split into the change in the mean salt content (an indicator of reformulation) and change in total volume sales using a decomposition formula described elsewhere [20,22]. In summary, the annual percentage change in the total volume of salt sold is equal to the sum of the annual percentage change in the mean salt content and the annual percentage change in the volume of foods sold. This was calculated at the total, category, and company level. As the data were not normally distributed (as assessed visually), differences in the sales-weighted mean salt content of each category in 2020 compared to 2015 were tested using a Kruskal–Wallis test, where brands were used as the unit of analysis and $p$-value cutoff set at 0.05.

## Sensitivity analysis

Euromonitor provides sales data for the leading brands, but each category is assigned an "others" volume that represents the small and local brands not covered in its brand list. We assumed that all products that were not matched with a brand were included under "others"

for the relevant category. In order to test what impact this had on the results, we conducted a sensitivity test by recalculating the sales-weighted mean salt content overall and by category with "others" excluded.

## Results

In 2020, we included 7,927 products for 393 brands manufactured by 105 individual companies. This remained relatively stable over time, with 9,492 products for 394 brands and 101 companies being included in 2015. A full table of descriptive statistics, including the number of brands and total number of products matched with both brands and "others," can be found in S1 Table.

The sales-weighted mean salt content of foods included in this study fell by 0.05 g/100 g, from 1.04 g/100 g in 2015 to 0.99 g/100 g in 2020, equivalent to −4.2%, although this decline was not statistically significant ($p$ = 0.13) (Table 2). There was a reduction of 10% or more in 2 out of 9 food categories: breakfast cereals and processed beans, potatoes, and vegetables. Meat, seafood, and alternatives, savoury snacks, and sauces gravy and condiments saw reductions of 9.2%, 8.0%, and 5.5%, respectively, while changes seen in bread (−1.6%) and butter and spreads (−2.2%) were small. There was a negligible increase in the sales-weighted mean salt content of ready meals, soup, and pizza (1.2%). None of the changes were statistically significant. When the sales-weighted mean salt content was calculated for each data source separately in 2018, there was no statistical difference in the results overall ($p$ = 0.40) or by category (see S2 Table), suggesting that changes data source over time have not influenced the trends in salt content.

Overall, the sales-weighted mean salt content (g/100 g) of categories remained consistent year to year (Fig 2), although the savoury snacks and sauces, gravy, and condiments categories saw larger changes in later years (2018 to 2020) compared to earlier years (2015 to 2017).

In 2020, the total volume of salt sold from these 9 categories was equivalent to 2.25 g per person per day and declined from 2.41 g person per day in 2015, a reduction of 0.16 g per person (6.7%, = 0.54). The majority of salt sold came from 3 categories: bread (24%); meat, seafood, and alternatives (19%); and cheese (12%) (Fig 3). There was no change in the proportions each category contributed over time.

**Table 2. Changes in sales-weighted mean salt content (g/100 g) by food category.**

| Category | 2015 | 2016 | 2017 | 2018 | 2019 | 2020 | Percentage change 2015–2020 (%) | Absolute change 2015–2020 (g/100 g) | Kruskal–Wallis test (p-value) |
|---|---|---|---|---|---|---|---|---|---|
| Bread | 0.95 | 0.94 | 0.93 | 0.92 | 0.93 | 0.93 | −1.6 | −0.02 | 0.96 |
| Breakfast cereals | 0.50 | 0.48 | 0.45 | 0.44 | 0.40 | 0.42 | −16.0 | −0.08 | 0.65 |
| Butter and spreads | 1.17 | 1.20 | 1.15 | 1.12 | 1.10 | 1.14 | −2.2 | −0.03 | 0.88 |
| Cheese | 1.71 | 1.68 | 1.66 | 1.59 | 1.64 | 1.64 | −3.9 | −0.07 | 0.32 |
| Meat, seafood, and alternatives | 1.46 | 1.46 | 1.44 | 1.34 | 1.32 | 1.32 | −9.2 | −0.13 | 0.56 |
| Processed beans, potatoes, and vegetables | 0.60 | 0.59 | 0.60 | 0.55 | 0.51 | 0.53 | −10.6 | −0.06 | 0.11 |
| Ready meals, soup, and pizza | 0.69 | 0.67 | 0.68 | 0.68 | 0.70 | 0.70 | 1.2 | 0.01 | 0.98 |
| Sauces, gravy, and condiments | 1.56 | 1.58 | 1.56 | 1.56 | 1.50 | 1.47 | −5.5 | −0.09 | 0.39 |
| Savoury snacks | 1.78 | 1.80 | 1.76 | 1.69 | 1.67 | 1.64 | −8.0 | −0.14 | 0.37 |
| **Total** | **1.04** | **1.04** | **1.03** | **1.00** | **0.99** | **0.99** | **−4.2** | **0.05** | **0.13** |

*Percentage change calculated before rounding to 2 decimal places.

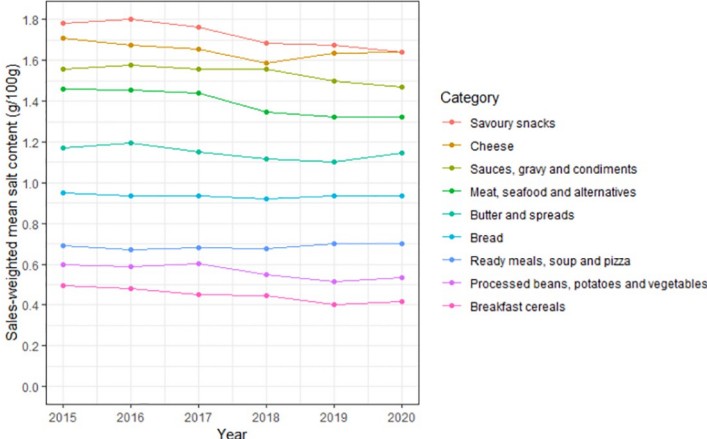

**Fig 2. Sales-weighted mean salt content (g/100 g) by category and year.**

The total volume of salt sold fell by 6.7% between 2015 and 2020 (represented by the black marker in Fig 4 below). The majority, or 63%, of this change was due to reductions in the mean salt content of brands (represented by the orange bar in Fig 4), with the remaining 37% due to a decline in the volume sales of products (represented by the grey bars in Fig 4). There were large reductions in the total volume of salt sold from breakfast cereals (−17.5%, $p = 0.88$); butter and spreads (−13.9%, $p = 0.89$); and processed beans, vegetables, and potatoes (−13.6%, $p = 0.31$), with the majority of this change being driven by reductions in the salt content of brands. Ready meals and cheese saw increases in the total volume of salt sold (+2.1%, $p = 0.89$), driven by higher brand sales, with reductions in the mean salt content of savoury snack brands being offset by increases in sales.

There was great heterogeneity between the changes in the mean salt content and sales between the top 5 companies in each category (companies that manufactured in more than 1 category were included separately for each category) (Fig 5). The total volume of salt sold by individual companies between 2015 and 2020 (represented by the black markers below) fell in 26 cases with increases in the remaining 19 companies. This company-level analysis allows for

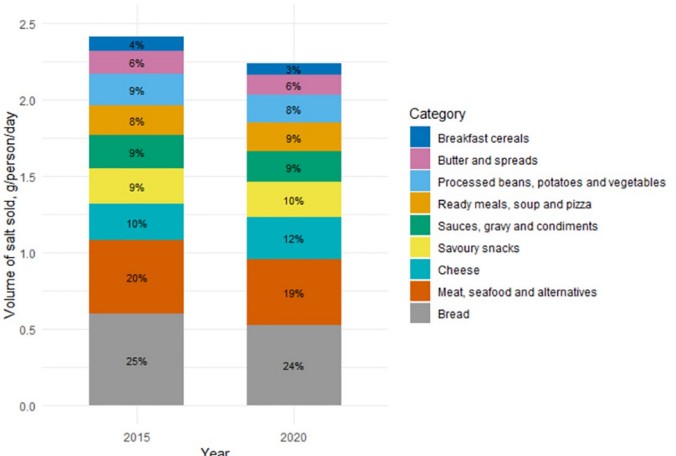

**Fig 3. Total volume of salt sold by proportion of each category, 2015 and 2020.**

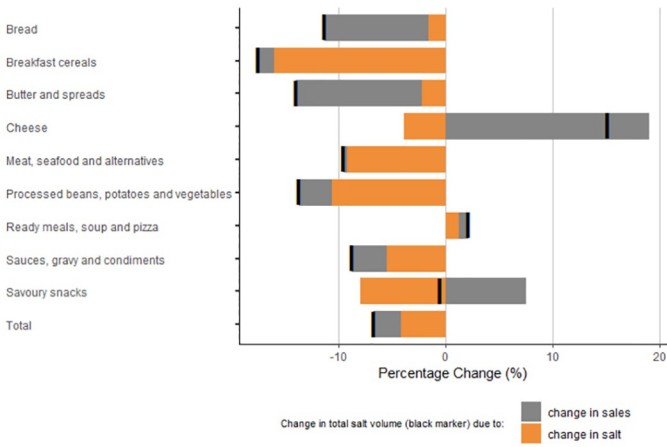

**Fig 4. Percentage change in total volume of salt sold due to change in sales and change in mean salt content, 2015–2020.**

best practices to be identified. For example, Tesco, Sainsbury's, and Morrisons all saw an overall reduction in the salt content of their meat, seafood, and alternative products, while Quaker and Tesco reduced the salt content of their breakfast cereals while increasing their product sales. However, the reductions in the salt content of ready meals, pizza, and soups were counterbalanced by increases in product sales, and there was great heterogeneity between company changes within the snacks and sauces categories too.

After excluding the "others" volume from the dataset and repeating the analysis, the overall sales-weighted mean salt content of foods was 0.95 g/100 g in 2020, compared to 0.98 g/100 g as presented in the main findings. The sales-weighted mean salt content and trends in change over time were similar across all categories (S3 Table), with the exception of 2 categories. The

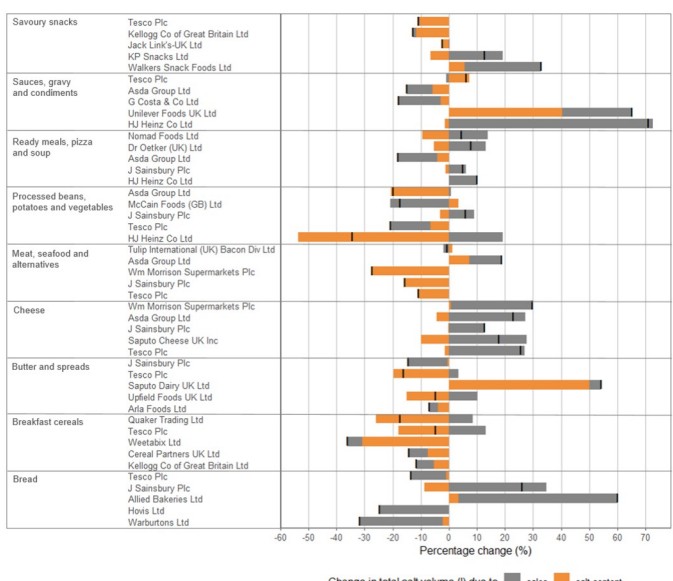

**Fig 5. Percentage change in total volume of salt sold by category and company, 2015–2020.** Sensitivity analysis results.

ready meals, soup, and pizza category saw salt content fall from 0.68 g/100 g in 2015 to 0.64 g/100 g in 2020 (−5.5%) in the sensitivity analysis, compared to +1.2% in the main findings, and the processed beans, potatoes, and vegetables category saw salt content fall from 0.54 g/100 g in 2015 to 0.44 g/100 g in 2020, a reduction of −17.6% compared to −10.6.5% in the main findings.

## Discussion

Between 2015 and 2020, the sales-weighted mean salt content of packaged foods fell by 0.05 g/100 g, or 4.2%, although this reduction was not statistically significant ($p = 0.13$). The total volume of salt sold from these foods fell from 2.41 g/person/day to 2.25 g, a reduction of 6.7% or 0.16 g. The majority (63%) of this reduction was driven by changes in the mean salt content of foods—either from reformulation of existing products, the launch of new low-sodium products, or removal of others from the marketplace—with the remainder due to falling sales. While there was a small overall decline in the volume of salt sold, there was also great heterogeneity between categories, and between companies within each category. The mean salt content was reduced by more than 10% in breakfast cereals and processed beans, vegetables, and potatoes. Bread contributed the greatest proportion to salt purchases; there was little reformulation seen over time, but sales decreased, leading to a small fall in the volume of salt sold. For cheese and ready meals, soup, and pizza, reductions in the mean salt content of products were counterbalanced by rising sales such that the total volume of salt sold increased over time.

### Comparison with other studies

To our knowledge, this is the first peer-reviewed study that has reported on company-level changes in the mean salt content of foods in the UK, although other studies have reported on category-level changes in the salt content of foods over time. A study by Ni Mhurchu and colleagues in 2011 examined the sodium content of 44,000 foods purchased by 21,000 UK households between 2008 and 2009 [23]. This study also reported that processed meat and bread were the largest contributors to sodium purchases, although they did not look at changes over time. A 2013 study by Eyles and colleagues found that the sodium content of 47,000 processed foods sold in 18,000 households fell by 7% between 2006 and 2011 [12]. This suggests that the rate of change we have seen in our study is in line with historic changes. Another smaller study looked at the salt content of bread in 2001 ($n = 40$) and 2011 ($n = 203$) and found that the salt content of these products fell by around 20% over time, with supermarket own products having a lower salt content than branded products [24]. This previous large reduction in the sodium content of bread may explain why our study found little further change in more recent years. However, the salt content of bread in 2020 was 0.93 g/100 g in 2020, above the category target of 0.90 g/100 g for 2017 [14] and with further progress needed to reach the target of 0.85 g/100 g for 2024 [11]. A study using household panel data between 2005 and 2011 found that meat and cheese, dairy, and fats were the main 2 categories contributing to salt intake at 23% and 19%, respectively, and were in line with the findings presented here, at 23% for meat and 17% for cheese and butter and spreads combined [25]. This study, and that by Gressier ad colleagues [13], supports this study's findings that that reformulation and industry behaviour were the main drivers of reductions in population salt intakes, as opposed to changes in consumer behaviour.

To our knowledge, there are no recent peer-reviewed studies that overlap with the time series presented here, although PHE has published a number of progress reports on the salt content of foods compared to the salt reduction targets. However, it is hard to make any direct comparisons between these reports and this study's findings, as the PHE salt reduction target

categories are so specific it is not possible to align categories. For example, in this study, we presented results for the "cheese" category, although data are available for hard cheese, processed cheese, and soft cheese. The PHE targets for cheese include "soft white cheese," "cottage cheese," "mozzarella," "blue cheese produced in the UK only," "cheese spreads," and "other processed cheese such as sliced and string cheese with emulsifiers" [11]. The sales data we used in this study were not granular enough to calculate the sales-weighted mean salt content of these specific PHE categories.

Results of a meta-analysis suggest that a 4.4-g/day reduction in salt intake leads to a 4.4-mm Hg fall in systolic blood pressure [6]. The change in total volume of salt sold observed in this study was −0.16 g/day; while any decline in salt intake is welcome, it seems unlikely that this change would have led to any substantial reductions in disease prevalence and mortality.

## Strengths and limitations of this study

By combining salt content data with sales data, we have been able to analyse the salt content of foods that have been sold, not just available products. This provides insight into how categories and companies have reduced the salt content of their products, and how sales—a marker of consumer behaviour—has changed over time.

The total volume of salt reported here initially appears low. For example, in 2018, we report 2.22 g salt/person/day compared to 8.4 g/person/day estimated in the urinary sodium excretion surveys. However, this discrepancy is due to a number of incremental factors. For example, we include only packaged foods sold through retail and exclude food sold for consumption outside the home in restaurants, cafes, and takeaways (30%; [26]), salt added at the table (20%; [27]), and sodium naturally occurring in food and drinks (10%; [27]). Furthermore, we were unable to analyse data from some categories (e.g., drinks, sweet bakery). With these adjustments, our estimates of salt per day are much more in line with the urinary sodium survey. It is also important to note that the sales data do not provide any split by demographics, and, therefore, any change in population composition (e.g., increase in proportion of older people) are not captured, making the daily per capita estimates presented here relatively crude. They include both adults and children of all ages and are therefore not directly comparable to the results of the urinary sodium figures, which are for adults aged 19 to 64 only.

The advantages and limitations of using Euromonitor sales data to analyse the nutrient content of foods over time have been outlined in detail in a previous study [20]. In summary, Euromonitor sales data avoid reliance on individual recall and underreporting in scan data [28,29], and it also allows for analysis of individual brands and companies. However, its limitations mean that table salt and salt sold from fast food are both excluded, making the total volume of salt sold likely to be a significant underestimation. Euromonitor's reporting of brand sales, and not individual product sales, is a major limitation and means that any heterogeneity that occurred between products that belong to the same brand would have been missed, given that the method used assumes all products under the same brand are sold equally. A previous study has examined this limitation in detail, concluding that while the overall findings are unlikely to be affected, there may be some significant misrepresentations at the brand and potentially company level [20]. Euromonitor is also limited in its granularity, meaning that is groups small and local brands for each category under the umbrella of "others," although the results of the sensitivity analysis show that this is unlikely to have a significant effect on the results overall but may have had a small effect on the ready meals, pizza, and soup and processed beans, potatoes, and vegetable categories. The lack of granularity in the sales data also means that these results are not to directly comparable with PHE's salt reduction targets, which have very narrow categories; therefore, this study is not a specific evaluation of the salt

reduction targets. However, the authors do have future research in this area planned, including modelling the targets' potential health impact.

There are several limitations to the nutrition composition database used in this study. The first is that we assumed that the products in the composition database were entirely representative of the brands that they matched with in the sales database. However, this may not be the case for several reasons: (1) any products that were out of stock on the day that the data were collected will not be included in the composition data; (2) it is possible that some brands have product variants, products that are only available in convenience stores, markets, or restaurants and cafes, which are not captured here; and (3) products sold exclusively in-store will not be captured by these data as these were only collected online. However, a previous validation study using the same foodDB database found that 85% of products sold in supermarket stores could be matched to those sold by the same retailer online, with nutrition information being identical [30].

Dried sauces and gravies were excluded from the dataset as no data were given to indicate whether the salt content values were as sold or as given. After investigation, most products had their own preparation instructions and attempts to devise a generic formula that could be used to convert all values over a certain threshold to as sold (as the sales data are given as sold) caused jumps in the category results across years; therefore, it was decided to exclude dried products. While the high salt content of dried products makes them a category of interest, results from the NDNS data show that the total sauces category represents 6% of salt intake overall, and it is not expected that their exclusion will have a large impact on the overall results.

## Policy implications

This analysis suggests that salt intake from processed foods purchased in retail settings fell by 6.7% between 2015 and 2020, although this decrease was not statistically significant. We highlight categories where progress is slow and the heterogeneity between companies within categories suggesting there is considerable scope for further reductions. In some categories, the salt content (e.g., bread and ready meals) has not changed in recent years. This is likely due to a combination of reasons: Many in the food industry cite the technical challenges to salt reduction, especially in products where it also acts as a preservative; there can be consumer resistance to low-salt varieties, although this can be overcome by gradual salt reduction; and there has been a shift in the efforts of both industry and policy makers towards sugar and calorie reduction instead. For savoury snacks and ready meals, reductions in salt content were counterbalanced by increases in sales. This highlights the need for policies that encompass consumer behaviour change and purchasing (such as reducing consumption of high-salt products and shifting towards low-salt varieties) as well as industry-led salt reduction targets and the launch of new low-salt varieties. The salt reduction programme in the UK was initially credited with such success because it included a public awareness campaign and strong political support [31], both of which have waned in recent years.

The specificity of the salt reduction targets means it is very hard to assess progress towards the targets using available data sources. Mandatory reporting of data to common standards would allow policymakers, academics, investors, and others to better track companies' progress towards public health goals, as has been called for in the National Food Strategy [32]. It would shine a spotlight on parts of the food system, which were leading or lagging behind policy targets, allow for a better understanding of how effective the current targets are at improving public health outcomes, and allow alternative measures to be

considered, such as mandated maximum salt targets (as in Argentina) and taxes on high-salt foods (as in Portugal) [33].

## Conclusions

In the period 2015 to 2020, there was a small decline in both the sales-weighted mean salt content of packaged grocery foods sold in the UK, and the total volume of salt sold from these foods, although these findings were not statistically significant. However, the heterogeneity between categories and between companies within categories suggests a more nuanced story, with progress stalling in some large and important categories such as bread and ready meals, inadequate action being taken by some companies in some categories and in some cases, reductions in the salt content of foods being counterbalanced by increased sales. Mandatory reporting of salt (and other nutrient) sales by large food companies would increase the transparency of how individual businesses are progressing towards the salt reduction targets and could therefore improve adherence.

## Supporting information

**S1 STROBE Checklist. STROBE statement.**
(DOCX)

**S1 Table. Number of brands and products included in the study by year and category.**
(PDF)

**S2 Table. Sales-weighted mean salt content by data source and category for 2018.**
(PDF)

**S3 Table. Results of sensitivity analysis—Sales-weighted mean salt content by category and year, with "others" removed.**
(PDF)

## Author Contributions

**Conceptualization:** Lauren K. Bandy, Susan A. Jebb, Peter Scarborough.

**Data curation:** Lauren K. Bandy, Sven Hollowell, Peter Scarborough.

**Formal analysis:** Lauren K. Bandy, Sven Hollowell, Peter Scarborough.

**Funding acquisition:** Lauren K. Bandy.

**Investigation:** Lauren K. Bandy, Susan A. Jebb, Peter Scarborough.

**Methodology:** Lauren K. Bandy, Sven Hollowell, Susan A. Jebb, Peter Scarborough.

**Project administration:** Lauren K. Bandy.

**Supervision:** Susan A. Jebb, Peter Scarborough.

**Validation:** Lauren K. Bandy, Peter Scarborough.

**Visualization:** Lauren K. Bandy, Peter Scarborough.

**Writing – original draft:** Lauren K. Bandy.

**Writing – review & editing:** Lauren K. Bandy, Susan A. Jebb, Peter Scarborough.

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
