## [Editor Report · Decision Letter 0]

18 Mar 2022

Dear Dr Bandy, 

Thank you for submitting your manuscript entitled "The salt content of packaged foods sold in the UK: A repeated cross-sectional study, 2015-2020" for consideration by PLOS Medicine.

Your manuscript has now been evaluated by the PLOS Medicine editorial staff and I am writing to let you know that we would like to send your submission out for external assessment.

However, we first need you to complete your submission by providing the metadata that is required for full assessment. To this end, please login to Editorial Manager where you will find the paper in the 'Submissions Needing Revisions' folder on your homepage. Please click 'Revise Submission' from the Action Links and complete all additional questions in the submission questionnaire.

Please re-submit your manuscript within two working days, i.e. by Mar 22 2022 11:59PM.

Once your full submission is complete, your paper will undergo a series of checks in preparation for external assessment. 

Kind regards,

Richard Turner, PhD

rturner@plos.org

---

## [Decision Letter · Decision Letter 1]

18 May 2022

Dear Dr. Bandy,

Thank you very much for submitting your manuscript "The salt content of packaged foods sold in the UK: A repeated cross-sectional study, 2015-2020" (PMEDICINE-D-22-00888R1) for consideration at PLOS Medicine. 

Your paper was discussed with an academic editor with relevant expertise and sent to independent reviewers, including a statistical reviewer. The reviews are appended at the bottom of this email and any accompanying reviewer attachments can be seen via the link below:

[LINK]

In light of these reviews, we will not be able to accept the manuscript for publication in the journal in its current form, but we would like to invite you to submit a revised version that addresses the reviewers' and editors' comments fully. You will appreciate that we cannot make a decision about publication until we have seen the revised manuscript and your responses, and we expect to seek re-review by one or more of the reviewers. 

We hope to receive your revised manuscript by Jun 08 2022 11:59PM. Please email us (plosmedicine@plos.org) if you have any questions or concerns.

Please let me know if you have any questions, and we look forward to receiving your revised manuscript. 

Sincerely,

Richard Turner PhD

Senior editor, PLOS Medicine

rturner@plos.org

Please adapt the title to better match journal style. We suggest: "Changes in the salt content of packaged foods sold in the United Kingdom during 2015-2020: A repeated cross-sectional study". 

Where available, please quote p values in the abstract. 

Please add a new final sentence to the "Methods and findings" subsection of your abstract, which should begin "Study limitations include ..." and should quote 2-3 of the study's main limitations. 

It is not clear that the sentence beginning at line 35 is a "conclusion", and we ask you either to remove this or to add "We suggest that ..." or similar. 

After the abstract, please add a new and accessible "author summary" section in non-identical style. You may find it helpful to consult one or two recent research papers published in PLOS Medicine to get a sense of the preferred style. 

At line 82, please remove "fellowship project".

Around line 88, please check that "Ascential" is spelt consistently. 

Early in the Methods section (main text), please state whether the study had a protocol or prespecified analysis plan and if so attach the relevant document as an attachment, referred to in the text. 

Please highlight analyses that were not prespecified. 

Noting the p values in table 1, for example, please note the statistical test used in a legend. 

Throughout the text, please amend reference call-outs to the following style: "... blood pressure [4,5]." (noting the absence of spaces within the square brackets). 

In the reference list, we suggest spelling out institutional author names such as "Strategy NF". 

Please include URLs for reports such as reference 10, where available, along with accessed dates. 

Please add a completed checklist for the most appropriate reporting guideline, e.g., STROBE, as an attachment, labelled "S1_STROBE_Checklist" or similar and referred to as such in the Methods section (main text). 

In the checklist, please refer to individual items by section (e.g., "Methods") and paragraph number, not by line or page numbers as these generally change in the event of publication. 

Comments from the reviewers:

*** Reviewer #1: 

I confine my remarks to statistical aspects of this paper. The general approach is fine, but I have some issues and suggestions before I can recommend publication.

p. 3 line 40 - I'm guessing this is per year, but you should state that.

 line 69 - similarly, I'm guessing this is per day, but you should say

p. 4 lines 86-94 Maybe put in a sentence such as "the method of reconciling these sources is discussed on p. 6" or something like that.

Table 1 - a minor point, and maybe I am wrong, but I'd consider ordering this table by %, rather than alphabetically. But I can also see reasons to leave it as is.

p. 7 line 155-6 I think you need to give the formula here, but I leave that to the editors' judgment. 

Fig 3, 4 - what do the black segments represent? Gray is sales, orange is salt. (On examination, kit looks like the black line is the total change. But 1) You should say this and 2) the black should be narrower, like a line rather than a box.

*** Reviewer #2: 

The topic of this paper is noteworthy, which investigates the salt content of packaged foods sold in the UK between 2015 and 2020. The authors report that there has been a steady decline in the salt content of packaged foods and total volume of salt sold between 2015 and 2020, with considerable differences between categories and companies. However, I have some concerns and suggestions for the authors, which I have detailed below. 

Major comments: 

1.The unit of analysis is the brand and the sales-weighted mean salt content per brand is the main primary outcome. There is a range of product for each brand. Within each brand, different sale quotas might be allocated to different products. Is it possible to segregate product sales? If such data is available, it would be better to explain why brand-level data should be used and to conduct sensitivity analysis (e.g. analyse the impact of individual product variation within each brand on the overall sales-weighted mean salt content of foods).

2.Page 7: dietary recommendations for adults and children are different. Total volume of salt sold in grams per person per day in this paper is averaged throughout the whole population, including adults and children. Meanwhile, the method used to determine it cannot be compared with 24h urine estimates. Therefore, more discussion is needed. 

3.Page 6: small and local brands are grouped under the umbrella term "others" and the remaining 40,003 products in the composition database are assigned to "others" within each category. It is unclear whether the sales volume data and nutrition information for the category "others" match exactly. A sensitivity analysis is required if this is not the case. 

4.Page 8: the P-values in Table 1 for changes in sales-weighted mean salt content (g/100g) of the 9 food categories are not statistically significant. In this case, reported deduction in the salt content of selected packaged foods might be caused by a random variation. As a result, conclusions can be misleading. It is necessary to provide clarifications or amend the conclusions so that they reflect the findings obtained.

5.Page 8: In Table 1, changes in sales-weighted mean salt content of 9 food categories are calculated between 2015 and 2020. As the yearly annual data is available, would be better to conduct an analysis that would account for / make use of all the data available.

Minor comments:

1.Page 3, line 69-71: citation 13 contains information that differs from that found in the reference paper (He FJ et al., 2014. Salt reduction in England from 2003 to 2011: its relationship to blood pressure, stroke, and ischaemic heart disease mortality). Please double-check it. 

2.Page 5: NDNS contributors in Table 1 needs a detailed explanation.

3.Black bars in Figure 3 (page 10) and Figure 4 (page 11) need an explanation. 

4.Page 8: The change in the sales-weighted mean salt content of bread (Table 1) between 2015 and 2020 is -0.01. When comparing the sales-weighted mean salt contents of bread in 2015 (0.93 ) and 2020 (0.93), the difference is 0. Therefore, it would be better to double check it and keep the right decimal. 

5.It would be better to discuss the implications of excluding dried sauces, gravy, and stock cubes. 

6.In the discussion section, would be better to mention Gressier et al. 2021's paper "Contribution of reformulation, product renewal, and changes in consumer behavior to the reduction of salt intakes in the UK population between 2008/2009 and 2016/2017". 

7.Page 12, line 217: it would be better to clarify that the sales-weighted mean salt content of packages foods fell by 0.07g per 100g. 

8.Page 13, line 223, please check that the comma is in the correct position.

*** Reviewer #3: 

This paper looks at the changes in salt content of foods between 2015 and 2020 in the UK. It's interesting work and relevant to understand how prior voluntary policies have managed to shape nutrient consumption (proxied through sales). I do have some comments and queries regarding the methods in particular which I think need further clarification. 

- Introduction - It would be good to have a more thorough overview of the work published investigating salt reformulation; where this paper fits in and what is the added value to the literature. I know of at least three papers that would be relevant for this study (at least for background) which are not mentioned

 o https://academic.oup.com/ajcn/article/114/3/1092/6272434?login=true

 o https://jech.bmj.com/content/73/9/881

 o https://onlinelibrary.wiley.com/doi/abs/10.1111/ecca.12192

- ln53 - clarify what you mean by the target - is it 'no more' than 6g/d?

- ln82 - is it important to note in methods that it is part of a fellowship project? That seems to be more appropriate to acknowledgement section together with fellowship grant reference

- How good is the match between the two data-sets on salt content of products (Edge by Ascential and foodDB)? - some validation information on this (validation) would have been useful to ensure that any changes observed are not due to using two different sources for 2015 and 2020.

- If I understand correctly there were 93,325 products matched into 395-416 brands. This means that each brand contains over 200 products and I question the use of average salt content because within such a large number of products salt content as well as popularity of individual products must vary (the very reason you do weighting at brand level) and I didn't see any clarification or justification about this. 

- Number of brands per food groups should be indicated in table 1. Table 1 should also have clearer indication to show what the last two column headings mean (i.e. % change between what, p-value for what?). Furthermore, you should use more than two decimals as two decimals show no change in salt content of bread for example.

- Line 179 - what do you mean by meaningful?

- Fig 3 & 4 - what do the solid black lines represent?

- Given that you have the whole series between 2015 and 2020 you could have also commented on the in-between years 

- The numbers in figure 3 does not match to the text explaining figure 3 so it makes a confusing read

- Table 1 shows that the decline in salt content was not significant at conventional statistical significance level and start the discussion states the decline as if it was. If you have chosen to ignore this for some reason, this should be carefully explained and why can you can still conclude this finding.

*** Reviewer #4: 

Bandy et al reported the changes in salt sold in the UK through retail between 2015-2020.

While the manuscript is generally well-written and the study well-conducted, there are several major issues which I would like to invite the authors to address:

1. A main limitation of the current study is the reliance of data collected from websites. There is no information provided in the manuscript, as to how the research team had assessed the accuracy of the data.

2. The authors selected nine categories, and claimed that eight of these represented ~73% of an adult's dietary sodium intake from processed foods. However products purchased are not only consumed by adults, but also children and adolescents.

3. Line 126: What is defined by Brand? e.g. For Kellogg's Special K, is Kellogg's or Special-K the Brand? This distinction is important given the sales data are weighted by "Brand", and if in the example above "Kellogg's" is identified as the Brand then the accuracy of the sales-weighted estimation will be too crude. This needs to be clarified. It is also surprising that only ~400 Brand exist in the UK market - does that mean each brand represent ~350 products (136k/395)?

4. Another issue is that the current study only examined packaged foods sold in the supermarket, as the authors themselves acknowledged.

5. The discussion is very brief. The authors should consider expanding it by including some or all of the following:

i) likely reasons for the small change observed, e.g. reluctance of the food industry, lack of consumer acceptance for lower Na products, technological challenges, etc.

ii) proportion of products meeting target over time

iii) likely impact of the current progress on diseases reduction (or lack thereof)

iv) strategies to enhance the compliance to the targets

v) complementary strategies for salt intake reduction in the population

There are also some minor issues that should be addressed:

1. Line 53: should this be < 6 g/day?

2. Line 112: do you mean savory or sweet biscuits? Also most of these quoted items are not low in sodium (as defined by the FSA traffic light label cut-offs), but only relatively lower in sodium. Excluding them may introduce error especially underestimation given some of these items were also included in the reformulation targets as far as I understand. "Making the categorization process more manageable" does not seem like a sound justification to me.

3. Regarding dried sauces/gravy etc. there could be some ground rules that would allow their inclusion. e.g. the difference between the dried vs made up version of these products would have huge differences in the sodium level, although this may require some manual checking.

4. The use of mean salt content for variations within a Brand may introduce error as their sales volumes are different.

5. Line 152: presenting the data as salt purchased per day per person and compared that across the year is likely misleading, given the population make up/demographics may have changed over the years. This should be discussed as a limitation.

6. No justification was provide for the use of the non-parametric Kruskal-Wallis test over its parametric counterpart. Was the normality of the data examined? If so, how? Also no p value cut-off was given.

7. Line 187: describing items which are no longer available for sale as "due to decrease in sales" is inaccurate.

8. Figure 3: No description on what the black bar means is given.

***

[LINK]

---

## [Decision Letter · Decision Letter 2]

25 Jul 2022

Dear Dr. Bandy,

Thank you very much for re-submitting your manuscript "Changes in the salt content of packaged foods sold in the United Kingdom: A repeated cross-sectional study during 2015-2020" (PMEDICINE-D-22-00888R2) for consideration at PLOS Medicine. 

Your paper was evaluated by an associate editor and discussed among all the editors here. It was also discussed with an academic editor with relevant expertise, and sent back to the reviewers. The reviews are appended at the bottom of this email and any accompanying reviewer attachments can be seen via the link below:

[LINK]

In light of these reviews, I am afraid that we will not be able to accept the manuscript for publication in the journal in its current form, but we would like to consider a revised version that fully addresses the reviewers' and editors' comments. We cannot make any decision about publication until we have seen the revised manuscript and your response, and we plan to seek re-review by one or more of the reviewers. 

We hope to receive your revised manuscript by Aug 15 2022 11:59PM. Please email us (plosmedicine@plos.org) if you have any questions or concerns.

We look forward to receiving your revised manuscript. 

Sincerely,

Callam Davidson, 

Associate Editor

PLOS Medicine

plosmedicine.org

The bullet at line 53 doesn’t seem to be complete.

S1 and S2 Tables are duplicated, and the STROBE checklist is missing.

Table S3 column headers appear to be truncated. 

Line 116: Please state in this subsection that your study does not have an associated protocol. 

Line 148: Please include the software manufacturers.

Line 323: Please confirm this remains accurate and add ‘to our knowledge’.

Lines 389-390: Please update to ‘This analysis suggests that salt intake from processed foods purchased in retail settings fell by 6.9% between 2015-2020, but this decrease was not statistically significant.’

Please include the date accessed for references 10 and 31. 

Comments from the reviewers:

Reviewer #1: The authors have addressed my concerns and I now recommend publication. 

Peter Flom

Reviewer #2: Overall, all of the comments have been addressed by the authors. I still doubt the validity of the datasets, e.g. duplicate counting of products sold in multiple retailers, which could affect the results. I leave that to the editors' judgment. A minor comment is that reference 14 should be checked properly. Instead of diastolic blood pressure, systolic blood pressure fell by 3.0mmHg.

Reviewer #3: All my comments have been responded to. I only have few minor comments to further make

Abstract

Some p-values missing

Can you provide p-values also for the change in total volume of salt (also in results section?) 

Author summary

Check the first bullet point in the second section - it does not seem to be a complete sentence

Last bullet point - this also needs a clarification that the biggest reductions were not statistically significant either

Results

The numbers cited in line 231-233 seem to be in reverse with reference to the year they are from. Sentences are missing the word 'products'

Discussion

First sentence - the p-value should be 0.23?

Linde 355 - what do you mean by 'industry behaviour' beyond reformulation as that is already mentioned. Consumer behaviour could also be explained a bit better- is it both buying less and buying products with lower salt content?

S3 table - column and row headings are not fully visible

S4 table - add also the Kruskal Wallis test as in table 2

Reviewer #4: I thank the authors for their attempt to address my concerns about their manuscript. I am confining my comments below to their responses to my original comments (Comment# corresponds to that in the original review)

Major:

1. What are the ~15% of offline store items not found on the website?

2. No further comment.

3. I felt that the authors did not adequately address this concern (raised by 3 reviewers). In Table S1, for example, there were 18 brands of bread in 2015, but that represents 1520 products. That means in the Euromonitor dataset there were ~90-100 items in each brand. The same is true for breakfast cereals and other categories. Using the authors' own argument in the response letter, that means Kellogg's Special K as a brand represent ~30 items in 2015? That does not seem to make sense. Even if this is true, it is reasonable to suspect that these items will be highly heterogenous, and thus using "brand" as a unit of analysis is quite likely to introduce sizeable errors in the estimation of sodium purchased.

4. How about convenience stores? That in my opinion would be a prime source of high sodium packaged foods.

5. I respectfully disagree with the authors' response. The current discussion is in my opinion inadequate and does not do justice to the data available, and I have in my original review provided some ideas of how the authors could better discuss their findings. 

Minor:

3. I disagree that it is not easy to create a ground rule to identify dried vs. as prepared. If nutrition information panel data are available, since the overall weight equal sum of CHO + protein + fat + fibre + water + (minerals), it would be quite easy to deduce the water content (and thus whether the nutrition label presents the dried vs. as prepared data). 

4. Please see Major3 above.

5. The authors did not completely address my concern. What I meant in my original comment is that the composition of the UK population had likely changed between 2015-2020. As an arbitrary example there may be more older adults or females in 2020 cf. 2015. This would likely affect the amount sodium purchased. Using the annual ONS statistics only allows the authors to correctly calculate the per person data, but not reflect this change in population composition.

6. This relates to line 147-148 in the original manuscript "Change in sales-weighted mean salt content was calculated between 2015 and 2020, with all products being included regardless of whether they were present in the market in 2015 or 2020.". From the wording it is implied that the value 62% was obtained from counting both items available at both 2015 and 2020 (and had little change in sales), as well as new items introduced in 2020; while the remaining 38% were items available in both 2015 and 2020 with a decline in sales, and those only available in 2015. It is "those only available in 2015" that I was referring to as "describing items which are no longer available for sale as "due to decrease in sales" is inaccurate.". The authors should better clarify this in text if my understanding is incorrect.

[LINK]

---

## [Decision Letter · Decision Letter 3]

25 Aug 2022

Dear Dr. Bandy,

Thank you very much for re-submitting your revised manuscript "Changes in the salt content of packaged foods sold in supermarkets in the United Kingdom: A repeated cross-sectional study during 2015-2020" (PMEDICINE-D-22-00888R3) for consideration at PLOS Medicine. 

Your paper was sent back to one reviewer and discussed by the editorial team and the academic editor. It was felt that there were remaining concerns regarding the analytical approach employed in the study, and we are still unable to accept the manuscript for publication in the journal until these are suitably addressed. The editorial comments are appended at the bottom of this email.

We hope to receive your revised manuscript by Sep 08 2022 11:59PM. Please email us (plosmedicine@plos.org) if you have any questions or concerns.

We look forward to receiving your revised manuscript. 

Sincerely,

Callam Davidson, 

PLOS Medicine

plosmedicine.org

Comments from the Academic Editor:

Please conduct and present an analysis that includes only the nutrition data for items with a full nutrition facts panel available, as the current approach has limited accuracy and could distort interpretation.

Additionally, please note explicitly (in both the Introduction and Discussion) that nutrition data was only available for a certain number of products and that this data was extrapolated to represent a greater number of total products, as this is important information for the reader to be aware of when interpreting the findings. 

Please update your title to ‘Changes in the salt content of packaged foods sold in supermarkets between 2015-2020 in the United Kingdom: A repeated cross-sectional study’

The limitations mentioned by the academic editor should also be included in the final sentence of your Abstract Methods and Findings and in the Author Summary.

Data Availability Statement: Typo in first sentence (‘This’).

Typo at line 289.

Comments from the reviewers:

Reviewer #4: The authors have addressed most of my comments adequately; and while I don't fully agree with the author's responses to some of my comments, I acknowledge that those are beyond the authors control, and hence with the acknowledgement of these as significant limitation of the study, the manuscript could be consider appropriate for publication.

[LINK]

---

## [Editor Report · Decision Letter 4]

5 Sep 2022

Dear Dr. Bandy,

Thank you very much for re-submitting your manuscript "Changes in the salt content of packaged foods sold in supermarkets between 2015-2020 in the United Kingdom: A repeated cross-sectional study" (PMEDICINE-D-22-00888R4) for review by PLOS Medicine.

I have discussed the paper with my colleagues and the academic editor. The academic editor still has concerns regarding the clarity of the methodology as it is currently presented. Providing the remaining comments (below) are suitably addressed in the next revision, we are planning to accept the paper for publication in the journal.

If you have any questions in the meantime, please contact me (cdavidson@plos.org) or the journal staff on plosmedicine@plos.org.  

We look forward to receiving the revised manuscript by Sep 12 2022 11:59PM.   

Sincerely,

Callam Davidson, 

Associate Editor 

PLOS Medicine

plosmedicine.org

Requests from Academic Editor:

At the beginning of your Methods section, please include the exact number of barcoded foods for which you collected the precise sodium content per 100 grams, and the exact number of additional foods that were imputed with these barcodes to produce the nutrition facts panel. 

Readers should be able to easily determine the total number of foods for which you collected all the nutrition facts panels, and the total number of foods for which sodium values were imputed (where exact sodium values were unavailable).

Editorial requests:

Lines 232/233: The word 'products' is missing.

Lines 242 and 246: Please address the typos on these lines.

---

## [Editor Report · Decision Letter 5]

21 Sep 2022

Dear Dr Bandy, 

On behalf of my colleagues and the Academic Editor, Professor Barry Popkin, I am pleased to inform you that we have agreed to publish your manuscript "Changes in the salt content of packaged foods sold in supermarkets between 2015-2020 in the United Kingdom: A repeated cross-sectional study" (PMEDICINE-D-22-00888R5) in PLOS Medicine.

To help us extend the reach of your research, please provide any Twitter handle(s) that would be appropriate to tag, including your own, your coauthors’, your institution, funder, or lab. Please email cdavidson@plos.org with any handles you wish to be included when we tweet this paper.

PRESS

Sincerely, 

Callam Davidson 

Associate Editor 

PLOS Medicine